# Evaluating Human–LLM Alignment Requires Transparent and Adaptable Statistical Guarantees

## Abstract

As Large Language Models (LLMs) become increasingly embedded in critical domains such as healthcare, education, and public services, ensuring their alignment with human values and intentions is of paramount importance. Misalignment in these contexts can lead to significant harm, underscoring the urgent need for rigorous, interpretable, and actionable evaluation methods. This position paper provides a critical examination of the current landscape of human–LLM alignment evaluation, with a particular focus on statistical guarantees in human annotation-based and LLM-based approaches. We identify key limitations in existing methodologies and **advocate for the development of more transparent, interpretable, and adaptable frameworks for alignment guarantees.** At the heart of our inquiry are two foundational questions: What constitutes a transparent foundation for alignment guarantees? And how can such guarantees be made operational and responsive to real-world conditions? We conclude by outlining future directions for designing alignment guarantee frameworks that are not only technically sound and transparent, but also socially attuned and practically adaptable.

## 1 Introduction

Large language models are increasingly integrated into real-world applications, from chat assistants to decision-support systems (OpenAI, 2024; Lin and Chen, 2023). However, ensuring that these models align with human values, preferences, and expectations has emerged as a central challenge (Dubois et al., 2023). This alignment—the degree to which LLM outputs match human expectations and values—represents both a technical and societal frontier in AI research.

Traditionally, the evaluation of LLM alignment has relied heavily on human judgments (Taori et al., 2023). While human-based annotation protocols offer direct insights into model-human agreement, they suffer from well-documented limitations, including subjectivity, limited diversity of annotators, poor inter-rater reliability, and high cost (Wu et al., 2023). Recent work has introduced more structured human evaluation protocols—such as pairwise comparisons and Elo-style rating systems—which offer greater statistical stability (Zheng et al., 2023; Dettmers et al., 2023), but do not resolve issues of scalability or systemic bias.

In parallel, the emergence of LLM-based evaluation has opened up promising new directions (Chiang and Lee, 2023). These approaches leverage LLMs themselves as evaluators, enabling scalable and cost-effective assessments across a range of tasks. However, they also come with significant limitations. Evaluator models are prone to positional and stylistic biases, self-enhancement effects, and susceptibility to subtle prompt manipulations (Wang et al., 2023a; Thakur et al., 2024). Moreover, as LLM-based evaluation inherits the limitations of its underlying models, it raises deep epistemological concerns about circularity, bias amplification, and the validity of using imperfect judges to evaluate other imperfect systems (Xiong et al., 2023).

To overcome these limitations, researchers have recently begun introducing statistical guarantees into alignment evaluation—borrowing tools from conformal prediction (Angelopoulos et al., 2022), PAC-style analysis (Jung et al., 2024), and risk calibration. These methods aim to formalize notions of alignment risk, abstention confidence, and human agreement, allowing for interpretable, probabilistic control over evaluation quality. However, despite these promising advances, current statistical approaches still face limitations in terms of generalization, robustness under distribution shift (Mohri and Hashimoto, 2024), interpretability for practitioners, and flexibility for different domains.

**This position paper advocates for a more transparent, interpretable, and adaptive statistical foundation for human–LLM alignment evaluation.** By transparent, we refer not only to the availability of formal guarantees, but also to the clarity with which their underlying components, assumptions, and limitations are communicated to users. A transparent framework should enable practitioners—and, where relevant, the public—to understand exactly what is being guaranteed (e.g., risk bounds, abstention criteria), under what conditions those guarantees hold (e.g., calibration set representativeness, model stability), and where the limits of validity lie (e.g., distribution shift, model uncertainty). By adaptive, we refer to the framework's capacity to accommodate task-specific requirements, user-defined risk tolerances, and domain variability. An adaptive statistical foundation should allow for dynamic calibration and parameterization (e.g., adjusting confidence thresholds or risk levels) to align with the practical demands and constraints of diverse deployment scenarios. Our central claim is that without transparent and adaptive statistical guarantees, alignment evaluations will remain fragmented, difficult to validate, and potentially misleading in real-world use. To structure our discussion, we pose two foundational questions:

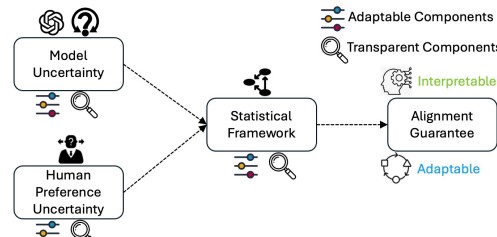

Figure 1: This figure illustrates a conceptual framework for generating statistical alignment guarantees that are both transparent and adaptable. The framework accounts for two primary sources of uncertainty: model uncertainty and human preference uncertainty. These uncertainties are modeled with both transparent components—such as calibration sets and empirical risk estimation—and adaptable elements, including task-specific uncertainty measures and tunable hyper-parameters. By integrating statistical tools with user-defined risk parameters, the framework yields formal guarantees on human–model agreement.

- **Transparency:** what constitutes a transparent and principled foundation for alignment guarantees?

- **Adaptability:** how can such guarantees be made operational—measurable, interpretable, and responsive to real-world deployment conditions?

We analyze existing evaluation methodologies (Sec. 2), review recent developments in statistical alignment guarantees (Sec. 3), and identify conceptual and practical gaps that persist. Finally, in Sec. 4, we argue that designing alignment guarantee frameworks with transparent and adaptable components is essential—not only for ensuring technical soundness, but also for fostering social trust, regulatory compliance, and safe deployment of generative models in high-stakes settings.

## 2 Existing evaluation methodologies

### 2.1 Human-based evaluation

Human–AI alignment evaluation has long been a central topic of study, early human evaluation frameworks adopted ordinal classification schemes, where annotators assigned responses to predefined quality levels. For example, Wang et al. (2022); Wu et al. (2023) used a four-point scale: acceptable, minor errors, major errors, and unacceptable. However, these categorical approaches suffer from substantial subjectivity, as evidenced by poor inter-annotator agreement in prior studies (Kalpathy-Cramer et al., 2016), highlighting the difficulty of applying rigid evaluation criteria to nuanced and context-dependent language outputs.

To mitigate these limitations, Taori et al. (2023) proposed a pairwise comparison protocol, where annotators judge which of two model responses is superior. This relative evaluation format reduces cognitive load and improves annotation consistency. Building on this, recent work such as Zheng et al. (2023); Dettmers et al. (2023) incorporates Elo rating systems, originally developed for ranking chess players, to dynamically assess model performance. In these systems, model scores are updated iteratively based on pairwise "wins" and "losses," enabling statistically robust comparisons across multiple LLMs.

More recently, human-based evaluation has advanced beyond static taxonomies and simple comparisons through the use of fine-grained rubrics and context-aware annotations. For instance, Fan et al. (2025) introduced SedarEval, a rubric-driven framework where task-specific rubrics are automatically constructed from prompts and refined through human judgment. In the safety domain, Xie et al. (2025) developed SORRY-bench, a large-scale corpus of over 7,000 human-annotated refusal cases, emphasizing diversity and inter-annotator agreement to assess LLM safety behavior. Arabzadeh and Clarke (2025) benchmarked LLM-generated judgments against expert relevance assessments in TREC RAG tasks, demonstrating the advantage of hybrid human–machine adjudication over fully automated metrics. Additionally, Yu et al. (2025a) proposed RPGBENCH, where humans interact with LLMs in role-playing scenarios to evaluate their behavioral consistency and narrative plausibility. Collectively, these works reflect a clear shift toward context-rich, trait-grounded human evaluation paradigms that more accurately capture the complexity of aligning LLMs with human expectations.

Through these progressive refinements in human evaluation protocols, the field has evolved toward more reliable and systematic assessment methodologies. However, several key challenges remain.

**Challenge (Subjectivity):** Human-based alignment evaluation is inherently subjective (Binns et al., 2018; Chang et al., 2024), often reflecting narrow cultural or demographic biases due to limited annotator diversity. This can skew alignment objectives and marginalize underrepresented perspectives. Moreover, a preference articulation gap—the mismatch between evaluators' intentions and how they score—introduces noise, as annotators may struggle to express preferences clearly or rationalize them inconsistently. Evolving social norms further complicate evaluation, making human preferences a moving target. Finally, conflicts between expert and general-user priorities—such as accuracy versus empathy—raise unresolved questions about whose preferences should define alignment.

**Challenge (Scalability):** Human evaluations face serious scalability constraints (Li et al., 2023). Recruiting and compensating annotators is costly, limiting coverage across use cases and depth in rare scenarios. As LLMs evolve rapidly, manual evaluations struggle to keep pace, often becoming outdated before deployment. The vast space of possible inputs makes exhaustive testing infeasible, especially for rare but critical failures. Additionally, annotator fatigue and limited domain expertise reduce evaluation quality over time, highlighting the need for more scalable, systematic alternatives.

## 2.2 LLM-based evaluation

While human evaluation provides high-quality insights, it faces well-known challenges in terms of scalability, efficiency, and cost. At the same time, the increasing fluency of LLMs has made it difficult for annotators to reliably distinguish between human- and model-generated text in open-ended tasks (Clark et al., 2021), prompting growing interest in using LLMs themselves as evaluators.

LLM-based evaluation approaches vary in design. Some extend traditional reference-based metrics by prompting LLMs to generate multiple paraphrased references, thereby expanding evaluation coverage (Tang et al., 2023). However, such methods still rely on at least one human-written reference. More recent reference-free approaches have emerged, where LLMs are prompted to directly assess response quality using task descriptions and evaluation rubrics (Liu et al., 2023; Fu et al., 2023; Chen et al., 2023; Chiang and Lee, 2023). These methods have been adapted to tasks such as summarization (Gao et al., 2023), code generation (Zhuo, 2023), open-ended QA (Bai et al., 2023), and dialogue evaluation (Lin and Chen, 2023), with prompt engineering enabling multi-dimensional assessments over quality, coherence, and factuality (Fu et al., 2023; Lin and Chen, 2023). Factuality remains a core focus of LLM-based evaluation. Studies have assessed factual correctness using both closed-source and open-source models (Min et al., 2023; Zha et al., 2023). Building on the success of human-based pairwise evaluation, models like GPT-4 have been used to conduct direct comparisons between candidate outputs (Dubois et al., 2023; Zheng et al., 2023).

Despite promising results, LLM-based evaluators exhibit notable biases. Wang et al. (2023a) observed positional bias, where models favor the first option regardless of content quality; mitigation strategies include candidate shuffling and chain-of-thought prompting. Wu and Aji (2023) reported that LLM judges often over-penalize grammatical issues and brevity while overlooking factual inaccuracies. To address this, a multi-dimensional Elo system has been proposed to separately score accuracy, helpfulness, and fluency. Zheng et al. (2023) also identified self-enhancement bias, where models tend to favor their own outputs. Remedies include randomized candidate positioning, exemplar conditioning, and reasoning-enhanced prompting.

Although LLMs like GPT-4 can match human raters in accuracy (Dubois et al., 2024; Li et al., 2024b), their use raises concerns about cost and bias. To improve efficiency and interpretability, researchers have explored judge model distillation (Kim et al., 2024; Zhu et al., 2023), peer review ensembles (Verga et al., 2024), and multi-agent debate systems (Chan et al., 2023). Still, most of these methods lack formal guarantees of reliability. Emerging studies further reveal that LLM judges are susceptible to cognitive and stylistic biases (Zeng et al., 2023; Koo et al., 2023; Panickssery et al., 2024), calling into question their robustness and generalizability. To address privacy and accessibility concerns associated with closed-source evaluators, Wang et al. (2023b) developed PandaLM, a fine-tuned LLaMA-7B model which achieves evaluation quality comparable to GPT-3.5 and GPT-4.

Recently, Wang et al. (2025b) proposed OpenForecast, where LLMs perform both forecasting and evaluation using retrieval-augmented prompts—eliminating the need for human-written references. Yu et al. (2025b) introduced xFinder, a unified interface for summarization and translation evaluation using instruction-tuned LLMs to assess fluency, adequacy, and factuality with improved human agreement. Badshah and Sajjad (2025) developed DAFE, a confidence-aware ensemble of multiple LLM judges. Cao et al. (2025) proposed the Multi-Agent LLM Judge, which assigns distinct personas to LLMs to support personalized, context-sensitive evaluations across traits such as coherence, specificity, and style. While such LLM-based evaluation methods represent substantial progress, several critical challenges remain for future investigation.

**Challenge (Echo Chamber Effects):** Using LLMs to evaluate other LLMs introduces circular reference problems that complicate alignment evaluation (Wataoka et al., 2024). When models evaluate outputs similar to what they might generate themselves, they often exhibit biases toward familiar patterns and approaches (Bommasani et al., 2023). The evaluating model itself may have alignment issues, creating a recursive problem of determining who evaluates the evaluators. Small changes in evaluation prompts can dramatically shift model judgments, raising questions about the stability of LLM-based evaluation methods. Judge models may show inconsistent calibration across different contexts, being overconfident in some domains and under-confident in others. Perhaps most concerning is the potential for bias amplification—when judge models with subtle biases are used to evaluate and train new models, these biases may be reinforced through successive iterations, creating problematic feedback loops in alignment systems that rely on model-based evaluation.

**Challenge (Inherent Uncertainty):** LLM-based alignment evaluation is fundamentally limited by the model's own epistemic and aleatoric uncertainties (Farquhar et al., 2024). As evaluators, LLMs generate preference judgments based on patterns learned from data, but lack true grounding or access to objective truth. This introduces epistemic uncertainty, especially in out-of-distribution or ambiguous cases where the model's internal representations are unreliable. In addition, aleatoric uncertainty arises when the evaluation instruction itself admits multiple reasonable interpretations, causing variability in outputs across different runs or prompts. Without principled mechanisms to quantify and communicate these uncertainties, model-generated evaluations may project a false sense of confidence, undermining their trustworthiness. This challenge is further exacerbated when such evaluations are used in downstream systems to guide training decisions, as unrecognized uncertainty can propagate misaligned updates and erode human trust in alignment processes.

## 3 Existing statistical guarantee for alignment

To address the limitations of prior human- and LLM-based methods, recent research has increasingly turned to enhancing LLMs with rigorous statistical guarantees aimed at controlling risk in high-stakes applications. Notable efforts include reducing hallucination rates in factual generation tasks (Yadkori et al., 2024; Mohri and Hashimoto, 2024) and controlling false discovery rates in medical decision-making (Gui et al., 2024). These approaches frequently leverage conformal methods (Angelopoulos

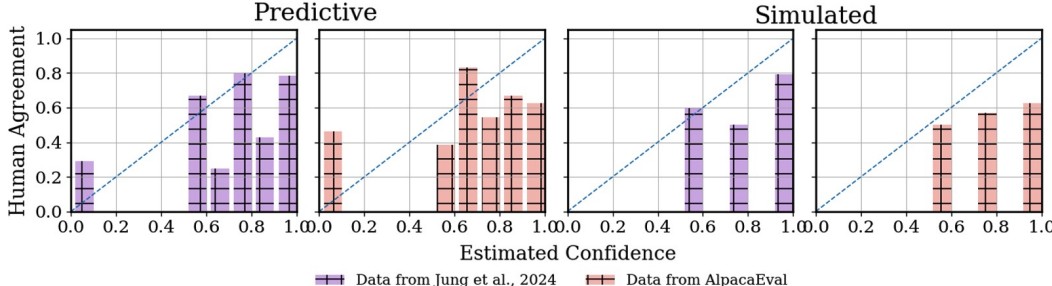

Figure 2: Reliability plot for confidence estimation methods (**left**: predictive probability measure; **right**: simulated annotators measure), using GPT-4 as judge on the data from Jung et al. (2024) (purple) and additional 500 records from AlpacaEval (orange) (Li et al., 2023). Horizontal axis represents the estimated LLM confidence, vertical axis represents the rate of human-LLM agreement, and dashed lines denote perfect calibration. More experimental details are given in Appendix A.

et al., 2022), which provide marginal control over prescribed risks. Complementary work has investigated fine-tuning objectives for LLMs to improve truthfulness (Kang et al., 2024; Tian et al., 2023) or to enable appropriate abstention when knowledge is insufficient (Zhang et al., 2024).

Notably, Yadkori et al. (2024) introduces a principled method to reduce hallucinations (enhance alignment) in LLMs by employing a conformal prediction-based abstention mechanism. The authors propose leveraging the LLM itself to evaluate the consistency among multiple responses generated for a given query, thereby measuring model uncertainty. Based on this uncertainty, their approach decides whether the model should respond or abstain, providing rigorous theoretical guarantees on limiting the rate of hallucinations. Mohri and Hashimoto (2024) also integrates conformal prediction into LLMs to ensure high-probability correctness guarantees for generated outputs. The authors conceptualize the correctness of an LLM's output as an uncertainty quantification problem, where each output corresponds to an entailment-based uncertainty set. By progressively "backing off" or making outputs less specific based on uncertainty estimates, the proposed method ensures that model outputs meet user-specified correctness levels with rigorous statistical guarantees.

Building on these foundations, Jung et al. (2024) extends them by developing an unsupervised confidence measure and establishing an exact upper bound on disagreement risk conditional on calibration set. Rather than issuing a decision unconditionally, the framework introduces a selective evaluation mechanism: the LLM makes a judgment only when it is sufficiently confident in its preference. This confidence is quantified by the confidence measure $\mathbb{C}_{LM}(x)$ for each input $x$, and a prediction is accepted if and only if the confidence exceeds a predefined threshold $\lambda$; otherwise, the model abstains.

Jung et al. (2024) frames the selection of $\lambda$ as a multiple hypothesis testing problem. Given access to a small calibration set of human preferences, they measure the empirical risk of disagreeing with humans when using threshold $\lambda$. Since the empirical risk follows a binomial distribution, they compute the exact $(1 - \delta)$ upper confidence bound of the risk. The risk tends to increase as $\lambda$ decreases, allowing to use fixed sequence testing (Bauer, 1991) to choose the threshold.

For a threshold chosen as above, and a selective evaluator operating based on the threshold, given a user-defined risk tolerance $\alpha$ and an error level $\delta$, they obtain the guarantee that:

$$\mathbb{P}(\text{human-model agreement}|\mathbb{C}_{LM}(x) \geq \lambda) \geq 1 - \alpha \qquad (1)$$

holds with probability at least $1 - \delta$. While this statistical guarantee represents a significant advancement, several challenges remain to be addressed in future work.

**Challenge (Confidence Measure):** While the simulated annotators confidence measure introduced by Jung et al. (2024) provides a promising approach to calibrating model judgments, its generalization capabilities across diverse tasks and domains remain uncertain. As LLMs are deployed in open-world environments, confidence scores derived from context-limited simulations may fail to capture the full variability of real-world queries. As shown in Fig. 2, the performance of the same confidence measure can vary substantially depending on the calibration set used. Moreover, the effectiveness of this measure in scenarios with highly technical or specialized content—where even human annotators might disagree significantly—requires further investigation. Future work should explore adaptive

confidence measures that dynamically adjust to task complexity and domain-specific characteristics, potentially incorporating domain knowledge and uncertainty quantification techniques.

**Challenge (Calibration Set):** The statistical guarantees provided by the framework rely critically on the assumption that the calibration set is representative of the distribution encountered during deployment (Malinin et al., 2021; Gui et al., 2024). In real-world scenarios, however, user queries may differ significantly from those in the calibration set—both in linguistic style and semantic content. This distributional shift jeopardizes the reliability of the estimated risk and its upper bound, leading to a potential mismatch between theoretical guarantees and practical performance. Future research should explore robust calibration methods that remain valid under distribution shifts, potentially incorporating concepts from domain adaptation, transfer learning, and human performance modeling to continuously update calibration parameters in response to evolving environments.

**Challenges in Transparency and Adaptability:** While the the previous works introduce promising statistical tools for alignment guarantees, the foundational underpinnings of these methods remain insufficiently examined. The effectiveness of current frameworks hinges on several assumptions that are often unverifiable or oversimplified in practice—such as the generalizability of confidence measures across domains, the monotonic behavior of empirical risk bounds, and the representativeness of calibration sets relative to deployment conditions (Angelopoulos and Bates, 2021). When these assumptions are violated—as is often the case in real-world settings—the guarantees provided become difficult to interpret, unreliable to uphold, and potentially misleading. This lack of clarity in the statistical foundation obscures the true meaning of alignment risk estimates and complicates their communication to developers, users, and regulators. Furthermore, in practice, different applications of LLMs impose distinct requirements on risk tolerance, abstention behavior, and evaluation criteria. Therefore, a key challenge lies in designing adaptive statistical guarantee frameworks that can be tuned to different tasks—whether through configurable risk parameters, dynamic confidence thresholds, or domain-specific calibration strategies. Without this adaptability, even well-calibrated guarantees risk being either too permissive in high-stakes settings or overly restrictive in low-stakes applications, ultimately limiting their real-world usability.

# 4 Future: A transparent and adaptable guarantee framework

To advance the interpretability and real-world applicability of human–LLM alignment guarantees, we advocate for the development of transparent and adaptable statistical frameworks. These directions aim not only to enhance the technical rigor of evaluation methods but also to ensure that alignment guarantees are trustworthy, interpretable, and practically deployable across diverse tasks and domains.

## 4.1 Transparency

Transparency is a prerequisite for trust—particularly in high-stakes applications where the consequences of model misalignment may be severe (Afroogh et al., 2024). While recent methods provide formal alignment guarantees, the internal mechanics, assumptions, and limitations of these frameworks are often opaque to both practitioners and end-users. We argue that statistical guarantees for alignment must not only be valid, but also interpretable and auditable. To achieve this, future frameworks should offer four essential pillars.

First, **explicit decomposition of guarantee components** is critical for demystifying the statistical machinery behind alignment evaluation. Each guarantee should be broken down into interpretable elements that explain its construction and function (Wei et al., 2024). This includes detailing how confidence scores are computed, how decision thresholds are selected to balance precision and coverage and how risk metrics—such as empirical disagreement rates or abstention-adjusted error bounds—are calculated. Furthermore, the abstention mechanism itself should be clearly explained, outlining when and why the model chooses to abstain, and what that implies for the overall evaluation coverage. Making these elements modular and transparent not only enhances trust but also facilitates debugging, tuning, and context-specific adaptation by downstream users (Wang et al., 2025a).

Second, any meaningful statistical guarantee must be accompanied by a **clear articulation of its assumptions and scope of validity**. Guarantees are only as strong as the premises on which they rest (Li et al., 2024a). Therefore, the framework must explicitly state the assumptions made about the data—such as the independence and identically distributed nature of calibration and test samples, the

representativeness of human preference annotations, or the reliability of confidence measures across input types. Additionally, assumptions about the model—such as the monotonicity of risk-confidence relationships or the correctness of label predictions—should be clearly noted. Where appropriate, the framework should specify for which domains, input styles, or task settings the guarantees are valid, and include warnings or diagnostics when these conditions are likely violated (e.g., due to distribution shift (Chopra et al., 2024), adversarial inputs (Chaudhary et al., 2025), or semantic ambiguity (Chaudhary et al., 2024)). This clarity is essential to avoid a false sense of security in settings where the guarantees may no longer be valid.

Third, **human-interpretable reporting** is indispensable for bridging the gap between technical precision and user-facing clarity. Statistical guarantees should be communicated in formats that facilitate understanding, decision-making, and trust (Wei et al., 2024). This involves translating formal quantities—such as confidence levels, coverage percentages, and upper bounds on alignment error—into natural language summaries that explain what these numbers mean in practice (Dubois et al., 2023; Lin and Chen, 2023). For example, rather than stating that "the upper bound on empirical disagreement is 0.05", the system could report that "the model is expected to agree with human preferences at least $95\%$ of the time when confident". Visualization tools such as risk-vs-coverage curves (Ao et al., 2023), abstention frequency histograms (Tayebati et al., 2025), or error calibration plots can further enhance comprehension. Additionally, contextual explanations that clarify why the model abstained or flagged uncertainty in a particular instance can empower users to make informed judgments, especially in domains such as medicine or law where interpretability is non-negotiable.

Finally, to support transparency at the ecosystem level, **auditable and reproducible evaluation** processes must be a cornerstone of any guarantee framework. This requires that the entire pipeline—from data collection and calibration to risk computation and threshold selection—be open to inspection, verification, and reuse. Practically, this means releasing detailed descriptions (or ideally open-source code) of how calibration datasets were sampled and processed (Yao et al., 2024), how risk statistics were computed (Tayebati et al., 2025), and how decision thresholds were derived (Sarmah et al., 2024). Evaluation tools should be modular and version-controlled, enabling consistent application across models and tasks while allowing traceability over time. Furthermore, when statistical claims are made—such as "the model meets a 95% alignment threshold"—external auditors should be able to reproduce the result from public artifacts. This level of transparency is essential not only for academic reproducibility, but also for regulatory oversight and responsible deployment in sensitive environments (Machado, 2025).

## 4.2 Adaptability

An adaptable statistical guarantee framework must be capable of responding to the diverse and evolving demands of real-world deployment contexts (Badawi et al., 2025). Unlike fixed, one-size-fits-all approaches, adaptability requires a framework that can be tuned to domain-specific constraints, task complexity, and operational realities. We identify the following characteristics of such a framework.

First, a statistically grounded guarantee framework should be **task-specific and configurable**. Alignment requirements and acceptable error rates vary substantially across use cases—what constitutes a tolerable mistake in a casual chatbot may be completely unacceptable in a clinical decision-support system (Kumar et al., 2025). Consequently, evaluation pipelines must offer control over key parameters such as abstention thresholds, acceptable risk bounds, and confidence thresholds. These parameters should not be hard-coded, but rather dynamically configurable based on the risk sensitivity of the task, its user base, or the deployment environment (Gallego, 2024). For example, an application in legal reasoning may demand a very low risk of misalignment with authoritative interpretations, justifying a high abstention rate; meanwhile, a creative writing assistant might prioritize broader coverage and fluency over strict alignment with normative content. An adaptable framework should allow such trade-offs to be explicitly set and monitored.

Beyond static configuration, alignment guarantee should also be **context-aware**—that is, sensitive to the semantic, social, and operational context in which the LLM is operating. Context-awareness includes the ability to incorporate auxiliary metadata, user roles (Sundaram et al., 2024), domain-specific knowledge (Zhao et al., 2024), or even prompt uncertainty (Martinson et al., 2025) into the evaluation logic. For instance, a system responding to novice users in educational settings might weight helpfulness and clarity more heavily than technical correctness, while the reverse may apply in scientific or engineering contexts. Guarantee criteria might also vary depending on input types

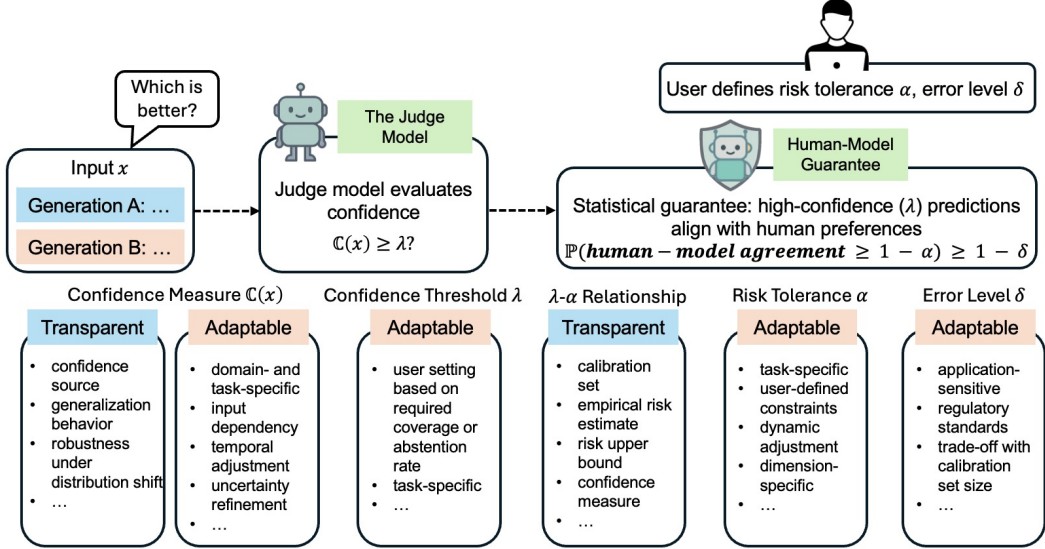

Figure 3: An example of the guarantee framework (Jung et al., 2024) with more transparent and adaptable components.

(e.g., structured queries vs. free-form dialogue) or user intent (e.g., exploratory vs. authoritative use). By embedding contextual signals into both risk estimation and abstention logic, the guarantee framework can become more aligned with the practical demands of different usage settings.

A major threat to the stability of statistical guarantees is distribution shift on calibration set. Therefore, adaptability also requires that the framework be **robust to domain shift**. Most existing methods assume that the calibration set used to construct statistical guarantees is representative of the deployment distribution—a condition that is rarely sustained in practice (Liu et al., 2024). A truly adaptable framework should include mechanisms to detect and respond to such shifts. This might involve monitoring model confidence drift, estimating divergence between calibration and live input distributions, or leveraging techniques from transfer learning and domain adaptation to re-calibrate guarantees in situ. Incorporating human-in-the-loop feedback, either through active learning (Goel et al., 2025) or post-deployment auditing (Cherian and Candès, 2024), can also help maintain the validity of guarantees over time. Without this robustness, statistical guarantees risk becoming brittle and misleading as models are deployed in new or evolving environments.

Finally, to support scalability and long-term usability, the guarantee framework should be **composable and extensible**. This means it should be modular in design, allowing components such as confidence estimation, calibration logic, and risk computation to be reused, replaced, or improved independently. Such modularity facilitates integration with different LLM architectures, evaluation settings, and interface modalities. It also enables researchers and practitioners to extend the framework—e.g., by incorporating new types of uncertainty quantification, social value priors, or hybrid human-AI judgment protocols—without requiring a complete system overhaul. A composable framework encourages experimentation and evolution, making it more likely to remain relevant as alignment research and model capabilities progress.

### 4.3 An example

Fig. 3 illustrates an enhanced version of the statistical guarantee framework introduced by Jung et al. (2024), enriched with explicit transparency and adaptability across its core components. The framework operates by assessing whether a judge model is confident enough—based on a confidence measure $\mathbb{C}(x)$—to make a reliable preference judgment between different generations. A prediction is only accepted if the confidence exceeds a threshold $\lambda$, thereby invoking a formal guarantee: with high probability $1 - \delta$, the probability of agreement with human preferences is at least $1 - \alpha$. This process ensures that the selected threshold satisfies the desired risk tolerance $\alpha$ under controlled sampling variability $\delta$, thus grounding the guarantee in observable empirical data.

This refined schematic highlights how each component of the evaluation pipeline—ranging from confidence estimation to risk quantification—can be made both transparent and adaptable. Trans-

parency is ensured through the explicit decomposition of evaluation elements, including the source and calibration of confidence scores, the construction of risk bounds, and the role of the empirical calibration set. Assumptions are made visible, such as the expected generalization behavior and the statistical relationship between confidence and error.

At the same time, adaptability is introduced through user-configurable parameters that tailor the framework to specific deployment scenarios. The confidence measure can be domain- and input-dependent, dynamically refined, and robust to distribution shifts. The confidence threshold $\lambda$ is adjustable based on desired abstention or coverage, while the risk tolerance $\alpha$ can be adjusted in accordance with task sensitivity. These dimensions collectively allow the framework to be customized for diverse applications—from high-stakes decision-making to exploratory human–AI interaction.

Together, these enhancements make the alignment guarantee framework not only more interpretable and auditable for developers and evaluators, but also significantly more practical for real-world, context-sensitive deployment.

## 5  Discussion

To address the challenges of subjectivity, inconsistency, and low inter-rater reliability in human evaluation (Binns et al., 2018; Chang et al., 2024), the proposed framework centers on the explicit decomposition of statistical guarantee components and human-interpretable reporting. This involves systematically modeling and exposing the uncertainties associated with both LLM predictions and human preference annotations—such as variability in annotator agreement or instability in model outputs. By breaking down the guarantee into its constituent parts (e.g., confidence scores, abstention thresholds, empirical risk bounds), the framework makes transparent what is being guaranteed, under which assumptions (e.g., representativeness of the calibration set), and where the limitations lie (e.g., under distribution shift or in edge cases). This transparency enhances interpretability not only for developers and model evaluators, but also for downstream stakeholders—particularly in sensitive domains where trust and accountability are essential. At the same time, the framework improves scalability (Li et al., 2023) by embedding statistical guarantees within LLM-based evaluation pipelines. Rather than relying on extensive human annotation for every deployment setting, it leverages a compact human-labeled calibration set to compute risk bounds, enabling consistent reuse of calibration, evaluation, and abstention logic across multiple tasks. This significantly reduces the dependence on costly, large-scale manual annotation.

In response to the limitations of LLM-based evaluation and the fragility of current statistical guarantee frameworks, the design incorporates transparent and adaptable components, selective evaluation, and robustness to calibration set shift (Malinin et al., 2021) as foundational principles. Given that LLM-based evaluators inevitably inherit biases—such as positional or stylistic preferences—from the underlying models they are built upon (Farquhar et al., 2024), the framework allows for user-defined risk tolerances and abstention criteria to adapt the evaluation process to specific task requirements, risk levels, and fairness considerations. Additionally, it introduces mechanisms for dynamic adjustment of guarantees based on the quality and characteristics of the calibration data, as well as detection of distributional drift between calibration and deployment inputs (Angelopoulos and Bates, 2021). This adaptive architecture ensures that alignment guarantees remain both valid and meaningful when applied to diverse real-world conditions, from high-stakes professional domains to more flexible consumer applications. Taken together, these elements form a robust, interpretable, and scalable foundation for alignment guarantee—capable of supporting both principled assessment and responsible deployment of LLMs.

## 6  Conclusion

In this position paper, we argued that ensuring reliable and trustworthy human–LLM alignment requires more than formal guarantees—it demands frameworks that are transparent in construction and adaptable to diverse deployment scenarios. We examined the limitations of current human- and LLM-based evaluation methodologies, as well as recent statistical guarantee approaches. To address these challenges, we argued for a principled framework that decomposes guarantees into modular components, clarifies assumptions, enables human-interpretable reporting, and supports task-specific configuration. By embedding transparency and adaptability as core design goals, we aim to bridge the gap between statistical rigor and real-world usability, advancing alignment evaluation methods that are not only technically sound but also socially accountable and practically implementable.

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

# A   Confidence measures

To calibrate when to trust each model's judgment, Jung et al. (2024) introduces simulated annotators confidence measure. This method simulates multiple human-like preferences to improve the calibration of the model's confidence estimation, ensuring that its evaluations are reliable and aligned with human preferences. For a given test instance $x$, and its associated preference labels $y \in \mathcal{Y}$ (e.g., $a_1$ or $a_2$ being preferred), the model calculates the probability $\mathbb{P}_{LM}(y|x)$ of each possible outcome (i.e., the preference label $y$). The model is given a few ($K$) examples of preferences provided by simulated annotators. These are used as context for the model's decision-making. The model is then prompted to predict a preference label based on this context, for a total of $N$ different simulations (i.e., simulating $N$ different annotators). Each simulated annotator produces a prediction for the preference label. Then, the **simulated annotators confidence measure** is defined as

$$\mathbb{C}_{LM}(x) = \max_y \frac{1}{N} \sum_{j=1}^{N} \mathbb{P}_{LM}(y|x; (x_{1,j}, y_{1,j}), \cdots, (x_{K,j}, y_{K,j})), \tag{2}$$

where $(x_{1,j}, y_{1,j}), \cdots, (x_{K,j}, y_{K,j})$ are $K$ examples of preferences provided by $j$-th simulated annotator. Specifically, the confidence measure is the average probability over all simulated annotators' predictions for the preference label. If the simulated annotators agree, the confidence measure is high; if they disagree, the confidence is lower.

The **predictive probability confidence measure** was proposed by Geifman and El-Yaniv (2017) in selective classification, it represents the probability assigned by LLM to its predicted label.

The code and original data for AlpacaEva were obtained from: `https://github.com/jaehunjung1/cascaded-selective-evaluation`.

In addition, we collected 500 supplementary records from AlpacaEval (Li et al., 2023), which have also been made available in the uploaded material.

