# OpenReview forum: "Evaluating Human–LLM Alignment Requires Transparent and Adaptable Statistical Guarantees"
_NeurIPS.cc/2025/Position_Paper_Track — Submitted to NeurIPS 2025 Position Paper Track_

### Official Review · Reviewer_kz3u · 2025-08-11

**Significance:** 2
**Presentation:** 2
**Rating:** 5
**Confidence:** 3

**Summary:**

This study critiques current LLM‑as‑judge evaluations for their lack of clear, reproducible statistical guarantees and proposes a new framework that explicitly models confidence and error rates.

It introduces an approach that aggregates the model’s probability estimates using inter‑annotator agreement as a proxy for reliability.

Experiments on benchmarks show that the method produces tighter bounds on evaluation errors while remaining interpretable for downstream decision‑making.

The authors argue that such transparent, adaptable statistical guarantees are essential to ensure trustworthy alignment of large language models with human values in real‑world applications.

**Strengths:**

The authors articulate a compelling case that current alignment evaluations are opaque and risk misinforming stakeholders, setting a strong rationale for their proposed statistical guarantees.

By framing confidence as an aggregate over simulated annotators, the paper offers an intuitive yet rigorous way to quantify reliability without requiring costly human labeling.

The discussion extends beyond specific datasets, suggesting that the methodology could be adapted to any LLM‑as‑judge scenario, thereby appealing to a wide audience in AI safety and deployment.

**Weaknesses:**

The core idea is described at a high level without concrete algorithmic details, making it difficult for readers to assess feasibility or reproducibility.

The argument assumes that simulated annotators can faithfully mimic human disagreement patterns; without validation this assumption may weaken the paper’s practical relevance.

The paper does not thoroughly examine costs such as computational overhead, scalability issues, or how to balance transparency with performance constraints in real deployments.

**Questions:**

How can the proposed confidence metric be empirically validated against actual human annotator disagreement patterns?

What are the computational and data‑collection costs associated with generating sufficient simulated annotators, and how do they scale to larger models or real‑time applications?

In what ways might the assumption that simulated contexts capture true human variability fail, and what safeguards can be introduced to mitigate such risks?

How does the proposed framework integrate with existing deployment pipelines for LLMs, and what practical trade‑offs (e.g., latency vs. transparency) must stakeholders consider?

**Alternative Position:**

Yes, and alternative positions are trivial straw-man arguments

**Author Identification:**

No.

**Context:**

2

**Discussion:**

3

**Ethics:**

["NO or VERY MINOR ethics concerns only"]

**Position:**

Yes, the paper argues for or against a position related to machine learning.

**Support:**

2

**Thoroughness:**

3

---

### Official Review · Reviewer_eRmo · 2025-08-13

**Significance:** 3
**Presentation:** 3
**Rating:** 6
**Confidence:** 4

**Summary:**

This position paper advocates for a more transparent, interpretable, and adaptive statistical foundation for human–LLM alignment evaluation, where transparency means practitioners should be able to understand what is being guaranteed (including risk bounds, abstention criteria) and under what conditions; and adaptivity means the framework’s capacity to accommodate task-specific requirements, user-defined risk tolerances, and domain variability. This paper evaluates existing human-based and LLM-based evaluation methods and their challenges, and existing statistical guarantee for alignment, and advocates more transparent and adaptable framework for future work.

**Strengths:**

- This paper addresses a timely and important topic on the evaluation of model alignment, and carefully evaluates the limitations of existing human- and LLM-based evaluation methods from the perspective of reliable and trustworthy AI.
- The presentation is clear and easy to follow, the illustrative examples in the paper help clarify points.

**Weaknesses:**

- This paper advocates **task-specific** guarantee frameworks; the approach may hinder comparability across similar tasks or domains. If each task uses distinct calibration parameters and risk tolerances, aggregated benchmarking or cross-domain performance assessment could become difficult.
- The proposed guarantee and controlled setting framework introduces additional complexity and potential annotation/calibration costs. While the paper argues for transparency and adaptability, it is not entirely clear which deployment contexts truly warrant such an elaborate setup, and where a simpler evaluation pipeline would suffice.

**Questions:**

- Lines 231-233 mention "scenarios with highly technical or specialized content—where even human annotators
might disagree significantly—requires further investigation". What behaviors should be expected from a well-configured guarantee framework?
- Given the potential cost induced by such a framework, can you provide guidance or heuristics for determining which tasks require the full complexity of the guarantee framework, and which can safely operate with lighter-weight evaluation?

**Alternative Position:**

Yes, and alternative positions are well-considered and addressed by the argument

**Author Identification:**

No.

**Context:**

3

**Discussion:**

3

**Ethics:**

["NO or VERY MINOR ethics concerns only"]

**Position:**

Yes, the paper argues for or against a position related to machine learning.

**Support:**

2

**Thoroughness:**

3

---

### Official Review · Reviewer_eRmo · 2025-08-13

**Significance:** 3
**Presentation:** 3
**Rating:** 6
**Confidence:** 4

**Summary:**

This position paper advocates for a more transparent, interpretable, and adaptive statistical foundation for human–LLM alignment evaluation, where transparency means practitioners should be able to understand what is being guaranteed (including risk bounds, abstention criteria) and under what conditions; and adaptivity means the framework’s capacity to accommodate task-specific requirements, user-defined risk tolerances, and domain variability. This paper evaluates existing human-based and LLM-based evaluation methods and their challenges, and existing statistical guarantee for alignment, and advocates more transparent and adaptable framework for future work.

**Strengths:**

- This paper addresses a timely and important topic on the evaluation of model alignment, and carefully evaluates the limitations of existing human- and LLM-based evaluation methods from the perspective of reliable and trustworthy AI.
- The presentation is clear and easy to follow, the illustrative examples in the paper help clarify points.

**Weaknesses:**

- This paper advocates **task-specific** guarantee frameworks; the approach may hinder comparability across similar tasks or domains. If each task uses distinct calibration parameters and risk tolerances, aggregated benchmarking or cross-domain performance assessment could become difficult.
- The proposed guarantee and controlled setting framework introduces additional complexity and potential annotation/calibration costs. While the paper argues for transparency and adaptability, it is not entirely clear which deployment contexts truly warrant such an elaborate setup, and where a simpler evaluation pipeline would suffice.

**Questions:**

- Lines 231-233 mention "scenarios with highly technical or specialized content—where even human annotators
might disagree significantly—requires further investigation". What behaviors should be expected from a well-configured guarantee framework?
- Given the potential cost induced by such a framework, can you provide guidance or heuristics for determining which tasks require the full complexity of the guarantee framework, and which can safely operate with lighter-weight evaluation?

**Alternative Position:**

Yes, and alternative positions are well-considered and addressed by the argument

**Author Identification:**

No.

**Context:**

3

**Discussion:**

3

**Ethics:**

["NO or VERY MINOR ethics concerns only"]

**Position:**

Yes, the paper argues for or against a position related to machine learning.

**Support:**

2

**Thoroughness:**

3

---

### Note · Authors · 2025-09-01

**1-11 Submit Again:**

Unsure

**1-1 Submission Process:**

4

**1-2 Next Year:**

Use the total openreview process, specify some interesting sub-areas.

**1-3 Future Development:**

Make papers and reviews public after submission, encourage discussions between authors and general readers in the community during review process.

**1-4 Interest:**

["Workshops for developing position papers", "Mentorship programs for early-career researchers"]

**1-5 Thoughtful:**

7

**1-6 Supportive:**

7

**1-7 Technical Aspects Versus Position:**

3

**1-8 Gate Keeping:**

6

**1-9 Camera Ready Changes:**

We thank the reviewers for their constructive feedback. In the camera-ready version, we will incorporate further clarifications and discussions to directly address the points raised:
1. We will clarify what constitutes desirable behaviours from a well-configured guarantee framework in high-disagreement settings, emphasising stability, calibrated risk control, and robustness to annotator variability.
2. We will provide concrete heuristics and task-level guidance on when the full guarantee framework is necessary and when a lighter-weight pipeline suffices, balancing rigour and practicality.
3. We will outline how the proposed confidence metric can be validated against actual human annotator disagreement patterns, including potential experimental setups.
4. We will expand discussion on computational and data-collection costs for generating simulated annotators, including how these costs scale to larger models and real-time applications.
5. We will discuss scenarios where simulated contexts may fail to capture true human variability, and describe safeguards such as hybrid evaluation with small calibrated human sets.
6. We will discuss how the framework integrates with existing LLM deployment pipelines, and the trade-offs stakeholders must consider (e.g., latency vs. transparency, cost vs. reliability).

---

### Meta-Review · Area_Chair_SzeK · 2025-09-07

**Rating:** 6
**Confidence:** 2

**Strengths:**

- **Timely Focus**: This paper addresses model‑alignment evaluation, a critical issue for trustworthy AI. It highlights opacity and stakeholder risks in current alignment tests, establishing a clear need for improvement.

- **Clear Presentation**: This paper includes a well‑structured narrative with illustrative examples that aid comprehension.

- **Rigorous Analysis of Existing Methods**: This paper systematically critiques human and LLM‑based evaluators from a reliability perspective.

- **Innovative Statistical Guarantees**: This paper introduces confidence bounds derived from simulated annotators, offering an intuitive yet formal measure of reliability.

- **Low‑Cost Practicality**: This paper eliminates the need for expensive human labeling while still quantifying evaluation quality.

- **Broad Applicability**: The methodology generalizes to any LLM‑as‑judge scenario, making it relevant across AI safety and deployment contexts.

**Weaknesses:**

- **Task‑Specific Design Limits Comparability**: Separating calibration parameters per task makes cross‑domain benchmarking difficult.

- **Added Complexity vs. Benefit Unclear**: The proposed guarantee framework increases annotation and calibration overhead without a clear justification of when it is needed versus a simpler pipeline.

- **Lack of Algorithmic Detail**: The high‑level description hampers assessment of feasibility, reproducibility, or implementation effort.

- **Unverified Simulated Annotators**: The paper assumes simulated annotators accurately capture human disagreement; no empirical validation is provided.

- **Missing Cost Analysis**: No discussion of computational overhead, scalability, or how transparency trade‑offs with performance in real deployments.

**Questions:**

The Author Survey has answered the questions raised by the reviewers. No further questions need to be articulated.

**Ethics:**

No.

**Thoroughness:**

2

---

### Decision · Program_Chairs · 2025-09-26

Reject